# Effects of Zn Contents on Microstructure and Mechanical Properties of Semisolid Rheo-Diecasting Al-xZn-2Mg-1.5Cu Alloys

**DOI:** 10.3390/ma15082873

**Published:** 2022-04-14

**Authors:** Saiheng Hou, Jian Feng, Song Chen, Fan Zhang, Daquan Li

**Affiliations:** 1State Key Laboratory of Nonferrous Metals and Processes, GRINM Group Co., Ltd., Beijing 100088, China; housaiheng0408@163.com (S.H.); chensong@grinm.com (S.C.); zhangfan@grinm.com (F.Z.); 2GRIMAT Engineering Institute Co., Ltd., Beijing 101407, China; 3General Research Institute for Nonferrous Metals, Beijing 100088, China

**Keywords:** Al-Zn-Mg-Cu alloys, rheo-diecasting, microstructure, mechanical properties

## Abstract

The microstructure and mechanical properties of semisolid rheo-diecasting Al-xZn-2Mg-1.5Cu alloys with different Zn contents were investigated by scanning electron microscopy (SEM), X-ray diffraction (XRD), hardness testing (HV) and room temperature tensile testing. Results show that the as-cast microstructure mainly consists of spherical α-Al and Mg(Al, Cu, Zn)_2_ phases. Furthermore, a small amounts of Al_7_Cu_2_Fe phases were also detected along the grain boundary. Increasing the Zn contents from 8–12%, the volume fraction of the Mg(Al, Cu, Zn)_2_ phases increases from 4.9–7.4%. After solution heat treatment at 470 °C for 8 h, most of the Mg(Al, Cu, Zn)_2_ dissolves into the α-Al matrix, while the Al_7_Cu_2_Fe phase keeps with remains. The yield strength linearly increases from 482 ± 5 MPa of 8% Zn to 529 ± 5 MPa of 12% Zn. While, the ultimate strength of 10% Zn is 584 ± 2 MPa, which is higher than that of the other two alloys. Moreover, the average elongation dramatically decreases from 13% for the 8% Zn alloy to 2% for the 12% Zn alloy.

## 1. Introduction

The Al-Zn-Mg-Cu alloys with high strength and toughness are widely used in the aerospace industry and are usually manufactured by wrought technique which is more expensive than the traditional casting methods [1,2]. However, Al-Zn-Mg-Cu alloys prepared by conventional casting methods are prone to form casting defects such as hot tearing and shrinkage porosity which will make the tensile properties worse [3,4]. The semisolid metal forming technology has many advantages and can directly cast Al-Zn-Mg-Cu wrought aluminum alloys [5,6]. The many advantages of semi-solid metal forming are due to its unique microstructure, a portion of the liquid phase that solidifies first is globular grains and suspended in the remaining liquid phase, which can be obtained by creating convection, such as vibration [7]. Such microstructure is more uniform during the die filling process than the fully liquid microstructure. Furthermore, there are better castability because the solidification shrinkage is smaller than fully liquid state. As so far, some literatures have reported the successful preparation of Al-Zn-Mg-Cu alloys by semi-solid metal forming methods [8,9,10,11,12,13,14]. Al-Zn-Mg-Cu alloys are usually manufactured by plastic deformation or rolling technique, and as-cast microstructure is not globular grains of the semi-solid metal but dendritic grains. After plastic deformation of the alloys, fibrous grain structure appeared and storage energy was generated, and recrystallization occurred in subsequent heat treatment such as solution heat treatment. Therefore, the microstructures of Al-Zn-Mg-Cu alloys prepared by the semi-solid metal forming technique and traditional plastic deformation technique are quite different.

The mechanical properties of Al-Zn-Mg-Cu alloys were mainly determined by the contents of Zn, Mg and Cu elements and heat treatment. The ultra-high strength of Al-Zn-Mg-Cu alloys was achieved by the precipitation strengthening mechanism [15,16]. The precipitation sequence of Al-Zn-Mg-Cu alloy is as follows [17,18,19,20,21,22]:GP Zone → η′ → η (MgZn_2_)

GP Zone and η′ are transition phases before the formation of equilibrium phase MgZn_2_, which are also mainly composed of the major alloy elements Zn and Mg. Liu et al. [23] reported that in the T6 state, the volume fraction and number density of the precipitates increased with increasing Zn content, and thus the mechanical properties of Al-Zn-Mg-Cu alloys increased. Iwamura et al. [24] found that adding higher Zn element to Al-Zn-Mg-Cu alloys has greater strengthening effect. Marlaud et al. [25] confirmed that Al-Zn-Mg-Cu alloys with higher Zn content have higher hardness, which is related to the increase of the volume fraction of precipitates. The increase of Zn content related to the increase of volume fraction and number density of precipitates. In order to enhance the strength of the Al-Zn-Mg-Cu alloys, the Zn content was gradually increased.

Mahathaninwong et al. [26] optimized the T6 heat treatment of the semisolid rheo-casting 7075 Al alloy, which is different from the same alloy prepared by plastic deformation method. The strength of the alloy in the T6 temper was then obtained, which was about 90 MPa lower than the wrought alloy target. Curle [27] prepared a semisolid rheo-diecasting 7075 alloy and obtained mechanical properties in T6 condition, which were not as high as those of the wrought alloy goal. The heat treatment and mechanical properties of Al-Zn-Mg-Cu alloys prepared by semisolid rheo-casting method and plastic deformation method are obviously different.

As the difference in as-fabricated microstructures led to the differences in microstructures and mechanical properties in the subsequent processing, this paper hence aims to investigate the evolution of microstructures and mechanical properties of Al-Zn-Mg-Cu alloys produced by semisolid rheo-diecasting process. Little study reported about the effects of Zn content on the microstructure and mechanical properties of semisolid rheo-diecasting Al-Zn-Mg-Cu alloys. In this study, we explored the influence of Zn content on microstructure and mechanical properties of semisolid rheo-diecasting Al-xZn-2Mg-1.5Cu alloys.

## 2. Materials and Methods

The experimental alloys were prepared with high-pure Al (99.99%), pure Zn (99.9%), pure Mg (99.9%), Al-50Cu and Al-5Ti-B master alloys (all compositions are in wt.% unless otherwise noted) supplied by the HeBei Lizhong Non-ferrous Metal Group Co., Ltd., Hebei, China. The chemical composition of the studied alloys, calibrated by ICP-AES, was shown in Table 1. The alloys were melted in a resistance furnace at 720 °C and degassed with argon using a rotary impeller. The molten melt was poured into a crucible with an internal diameter of 78 mm and a height of 210 mm at 650 °C. The semisolid slurry was prepared by the Swirled Equilibrium Enthalpy Device (SEED, developed by Alcan, Canada) process at an eccentric rotation speed of 180 rpm (swirling). The prepared semisolid slurry was transferred to the Buhler 340-ton die-casting machine with a boost pressure of 95 MPa and a filling speed of 0.2 m/s. The casting has a diameter of about 45 mm and a length of 100 mm. Subsequently, the as-cast samples were solution heat treatment at 470 °C for 8 h (as-solution) and water quenched, then aging at 120 °C for different hours.

The microstructure observation and local chemical analysis of the unetched samples were investigated using a scanning electron microscope (SEM, JSM-7610F-JOEL, Tokyo, Japan) equipped with an Oxford energy-dispersive X-ray spectroscope (EDS, Aztec, Oxford Instruments, Oxford, UK) working at an accelerating voltage of 20 kV. The information of grain size and shape factor of the spherical α-Al and the volume fraction of the non-equilibrium phases were accurately analyzed by an image processing and analysis software (IPP, Image-Pro Plus 6.0 software developed by Media Cybernetics, Rockville, MD, USA). The grain size was calculated as: d = (4 A/π)^1/2^(where A represents the area of a grain). The shape factor was calculated as: F = 4π A/P^2^ (where P represents the perimeter of a grain). The X-ray diffraction (XRD) patterns of the samples were examined using the Bruker D8 Advance diffractometer (Bruker, Karlsruhe, Germany) operating at 40 kV and 40 mA with Cu Kα radiation in the range of 10–90° at a scanning speed of 8°/min. The microhardness was measured by a Vickers-hardness tester (HV-30, Shanghai, China) according to the Vicker hardness test standard (GB/T4340.1-1999). For each composition 5 tests were performed and the average value has been calculated. Tensile testing was conducted at room temperature on a tensile tester (E45.305, Shenzhen, China) with an initial strain rate of 1.6 × 10^−4^ s^−1^. Dog-bone shaped tensile test bars with a gauge diameter of 5 mm and gauge length of 25 mm were used in the tensile test.

## 3. Results

### 3.1. As-Cast and As-Solution Microstructure

Figure 1 shows the as-cast microstructure of the alloys which mainly consists of spherical α-Al instead of dendritic grain and white non-equilibrium phases on the grain boundaries. The average grain size and shape factor statistics of α-Al were shown in Figure 2a. It can be seen that the Zn content has little effect on the grain size and shape factor of the alloys, which was about 65 µm and 0.75, respectively. During solidification, the external conditions of the melt were the same and only the Zn content of the alloys was different, so α-Al cannot be refined by increasing the degree of undercooling or heterogeneous nucleation [28,29]. Figure 2b shows the statistical results of the volume fraction of the non-equilibrium phases on the grain boundaries in the as-cast state. With the increasing Zn content, the area fraction increases from 4.9% of 8 Zn, 5.4% of 10 Zn to 7.4% of 12 Zn. The content of the Zn element increases, and the content of the non-equilibrium phases precipitated during the solidification also increases.

In order to dissolve the non-equilibrium phases into the α-Al matrix, so that the nano-sized strengthening phase precipitated in α-Al matrix during aging, the as-cast alloys were subjected to solution heat treatment at 470 °C for 8 h. The SEM images after solution heat treatment were shown in Figure 3. Comparing with Figure 1, it can be seen that most of the white non-equilibrium phases on the grain boundaries dissolves into the α-Al matrix. Furthermore, the phases of as-cast and as-solution microstructures were analyzed by XRD, and the results were shown in Figure 4. It is clearly seen that the as-cast microstructure with different Zn contents mainly consists of α-Al and MgZn_2_. Many earlier studies [30,31,32,33,34,35] reported that the different non-equilibrium phases of Al-Zn-Mg-Cu alloys like MgZn_2_, Al_2_CuMg, Al_2_Mg_3_Zn_3_ and Al_7_Cu_2_Fe. The type of non-equilibrium phases is mainly determined by the Zn, Mg and Cu contents. It has been found that a relatively higher Zn content with lower Cu content contribute to the formation of MgZn_2_ instead of Al_2_CuMg [36,37]. The Zn content of the studied alloys is relatively high and the Cu content is only 1.5 wt.%, so the main non-equilibrium phases is MgZn_2_. Some investigators have also reported that the phase is MgZn_2_ in alloys with higher Zn content and lower Cu content [38,39]. In other words, the type of the non-equilibrium phases of the as-cast alloys were not affected by the Zn content. After solution heat treatment, most of the diffraction peaks of the MgZn_2_ phase disappears, indicating that most of the MgZn_2_ dissolves into the α-Al matrix. However, small amounts of diffraction peaks of MgZn_2_ phase were still detected, meaning that the MgZn_2_ did not completely dissolves into the α-Al matrix. Combining with XRD and SEM microstructure characterization, the non-equilibrium phases in the alloys were confirmed. SEM images of as-cast and as-solution alloys were shown in Figure 5. The composition of the phases present in Figure 5 was shown in Table 2. It can be seen that the atom ratio of Mg to the sum of Al, Cu and Zn is about 1:2. Al and Cu replace Zn of binary η (MgZn_2_), which can be described as Mg(Al, Cu, Zn)_2_ [40] and the composition of Mg(Al, Cu, Zn)_2_ has been found in Al-Zn-Mg-Cu alloys [41]. The deep gray phase in the as-cast microstructure contains Cu and Fe, and the composition is close to Al_7_Cu_2_Fe phase (spot 3 and 5 in Figure 5). It did not dissolve into the α-Al matrix after solution heat treatment [31,32,35]. However, the content of Al_7_Cu_2_Fe phase was very small, so there was no diffraction peaks were observed in Figure 4.

### 3.2. Mechanical Properties

Figure 6 shows the change in hardness of the studied alloys aging at 120 °C for different time, the first data point on each curve is the as-solution hardness. The as-solution hardness value in the 8 Zn, 10 Zn and 12 Zn alloys was 69, 88 and 104 HV, respectively. With the increase of Zn content, the supersaturation degree in the α-Al matrix increased after solution heat treatment (470 °C 8 h). Due to the effect of solid solution strengthening, the hardness increased with the increase of Zn content. It can be seen that the hardness increases rapidly with time and reaches the maximum value at about 24 h (T6 state). This is attributed to the gradual precipitation of GP zone and η′ (MgZn_2_) phase in the α-Al matrix, which can effectively prevent the movement of dislocations [42,43,44,45,46,47]. As the Zn content increases, the hardness of the alloys also increases during aging process. The increase of Zn content increases the supersaturation degree of α-Al after solution heat treatment (470 °C 8 h), which leads to the precipitation of more GP zone and η′ (MgZn_2_) phase during aging heat treatment [23,25]. Figure 7 illustrates the typical stress-strain curve of the studied alloys in T6 state (470 °C 8 h + 120 °C 24 h). The yield strength linearly increases from 482 ± 5 MPa of the 8 Zn alloy to 529 ± 5 MPa of the 12 Zn alloy. While, the ultimate strength of the 10 Zn alloy is 584 ± 2 MPa, which is higher than that of the other two alloys. Moreover, the average elongation dramatically decreases from 13% for the 8 Zn alloy to 2% for the 12 Zn alloy. After aging heat treatment at 120 °C for 24 h, the precipitation phases of Al-Zn-Mg-Cu alloy are mainly GP zone and η′ (MgZn_2_) phase [48,49]. When the dislocation moves in the α-Al and encounters the precipitates, it is required to cut the precipitates, which hinders the movement of the dislocation [50]. Therefore, the yield strength of the alloys increases. The higher the Zn content in alloy, the more precipitation in the T6 state alloys. From this analysis, the yield strength of the alloys was shown to increase with the increasing Zn content. Al-Zn-Mg-Cu alloys increase the strength but reduce the elongation by T6 treatment [51]. However, this yield strength is lower than the same Al-Zn-Mg-Cu alloy prepared by the plastic deformation method [23,52]. Compared with the Al-Zn-Mg-Cu alloys that underwent plastic deformation, the deformation strengthening effect of the studied alloys is very small and thus the yield strength is lower. The 12 Zn alloy contains the highest Zn content in the studied alloys, and the as-cast microstructure contains the most Mg(Al, Cu, Zn)_2_ phases (as shown in Figure 2b). The three alloys underwent the same solution heat treatment (470 °C 8 h), so the 12 Zn alloy may have more non-equilibrium phases that did not dissolved into the α-Al matrix. These undissolved Mg(Al, Cu, Zn)_2_ phases are the area of the stress concentration and promote the crack initiation and propagation, which reduces not only the plasticity, but also the ultimate strength of the alloys. Therefore, the tensile strength of 12 Zn alloy is slightly lower than that of 10 Zn alloy.

### 3.3. Fracture Morphology

Typical fracture morphology of the alloys under T6 state were shown in Figure 8. The fracture surfaces of the three alloys have a “rock candy” appearance which is a typical intergranular fracture. However, there are also a small amount of fine dimples, and the proportion of dimples tends to decrease with higher Zn content. A larger number of shear zones and fine dimples could be seen in the 8 Zn alloy (Figure 8d). Therefore, the 8 Zn alloy has the highest elongation. It is in good accordance with the experimental results of Figure 7. As in the previous discussion, the precipitation phases of the studied alloys under T6 state are mainly GP zone and η′ (MgZn_2_) phase, and dislocations can cut through it when encountered in the slip process. It led to the stress concentration and final intergranular fracture behavior [50]. In addition, the intragranular strength is enhanced by the precipitation phases, which also contributes to intergranular fracture [53]. During the tensile deformation process at room temperature, when the dislocation meets the coarse precipitates, it will accumulate and form microcracks around the precipitates. As a result of gliding, the microcracks gradually grow and connect with each other, eventually forming transgranular dimples [50,54].

## 4. Discussion

According to the results of the above microstructure observations, as the Zn content increases from 8–12%, the volume fraction of the non-equilibrium phases Mg(Al, Cu, Zn)_2_ in the as-cast alloys increase from 4.9–7.4% (Figure 2b). After solution heat treatment (470 °C 8 h), most of the non-equilibrium phases Mg(Al, Cu, Zn)_2_ of the three alloys dissolve into the α-Al matrix (Figure 3), indicating that with the increase of Zn content, more major elements (Zn, Mg and Cu) dissolve into the α-Al matrix, that is, the supersaturation in the α-Al matrix is greater. In the T6 condition (120 °C 24 h), the aging precipitates in the studied alloys are mainly GP zone and η′ (MgZn_2_) phase [48,49]. Therefore, when the dislocation encounters the GP zone and the finer η′ phase during slipping, it has to cut through them to continue moving, and then accumulate at the grain boundary. The dislocation cuts the GP zone and the finer η′ phase can improve the strength of the alloys, which can be expressed by the following formula [55]:∆σ = c_1_f^m^r^n^
where c_1_, m and n are constant; f is the volume fraction of the precipitates, r is the radius of the precipitates.

With the increase of Zn content, the volume fraction and number density of the precipitates increase [23,25], and according to the above formula, the yield strength increases with the increase of Zn content. When the dislocations cut through the precipitates, they aggregate at the grain boundaries, causing stress concentration and promoting intergranular fracture [50]. However, when the dislocation encounters the coarse precipitation η′ phase during slipping, it can only bypass it, which increases the resistance of the dislocation movement, thereby improving the strength of the alloys. The strengthening effect can be expressed by the following formula [56]:∆σ = c_2_f^1/2^r^−1^
where c_2_ is constant; f is the volume fraction of the precipitates, r is the radius of the precipitates.

When the dislocations bypass the precipitates, they will accumulate and form microcracks around the precipitates, and then the microcracks grow and connect each other to form transgranular dimples [50,54]. As discussed earlier, as the Zn content increases, the supersaturation degree of the alloy increases, and thus the driving force for aging precipitation also increases resulting in more coarse precipitates in the 8 Zn alloy. Therefore, the number of dimples in the 8 Zn alloy is the largest indicating that the plasticity of the alloy is the best (Figure 8). Furthermore, the strength and plasticity of materials have always been in a contradictory relationship [57]. With the increase of the volume fraction and number density of the precipitates, the resistance of dislocation motion becomes larger, and the yield strength of the alloy increases. However, plastic deformation is a process of dislocation multiplication and movement, and the resistance of dislocation movement increases, making it difficult to continue plastic deformation, so the plasticity of the alloy deteriorates. Therefore, when the Zn content is greater than 8%, the average elongation of the alloy decreases sharply, from 13% of the 8 Zn alloy to 2% of the 12 Zn alloy. In this study, the mechanical properties of the 8 Zn alloy are optimal, the yield strength is 482 ± 5 MPa, the tensile strength is 564 ± 3 MPa, and the elongation is 13 ± 1%. When the Zn content is 8%, the Al-xZn-2Mg-1.5Cu alloy prepared by semisolid rheo-diecasting has the feasibility of using.

## 5. Conclusions

In this paper, the effects of Zn content on the microstructure and tensile properties of Al-xZn-2Mg-1.5Cu alloys prepared by semisolid rheo-diecasting were systematically studied. The main results obtained from the experiments are as follows:

There are globular α-Al grains in the as-cast microstructure of three Al-xZn-2Mg-1.5Cu alloys with different Zn contents prepared by semisolid rheo-diecasting, and the non-equilibrium phases on the grain boundary are mainly Mg(Al, Cu, Zn)_2_. As the Zn content increases from 8% to 12%, the volume fraction of the Mg(Al, Cu, Zn)_2_ phase increases from 4.9% to 7.4%. After solution heat treatment (470 °C 8 h), most of the non-equilibrium phases Mg(Al, Cu, Zn)_2_ of the studied alloys dissolve into the α-Al matrix indicating that with the increase of Zn content, more elements dissolve into the α-Al matrix, that is, the supersaturation degree of the α-Al increases.In the T6 condition (120 °C 24 h), the yield strength of the alloy increases with increasing Zn content, but the elongation decreases sharply, from 13 ± 1% for the 8 Zn alloy to 2 ± 1% for the 12 Zn alloy.The fracture surfaces of the studied alloys in the T6 state are “rock candy” appearance, which is a typical intergranular fracture. In addition, the fracture surfaces of the three alloys contain a small amount of fine dimples, and the 8 Zn alloy has the largest number of dimples confirming its better plasticity.

Based on the research results of this paper, a kind of Al-Zn-Mg-Cu alloy composition prepared by semisolid rheo-diecasting with good tensile properties, namely Al-8Zn-2Mg-1.5Cu, was obtained. However, this tensile properties is lower than that of the same alloy prepared by the plastic deformation method. This may be related to the fibrous grain structure produced by the plastic deformation process, and the dislocation density after deformation is higher than that of the alloys studied in this paper. In the future, employing TEM analyze the precipitation of the Al-Zn-Mg-Cu alloys prepared by semisolid rheo-diecasting during the aging process will be particularly important, such as the volume fraction, size distribution and precipitation location etc.

## Figures and Tables

**Figure 1 materials-15-02873-f001:**
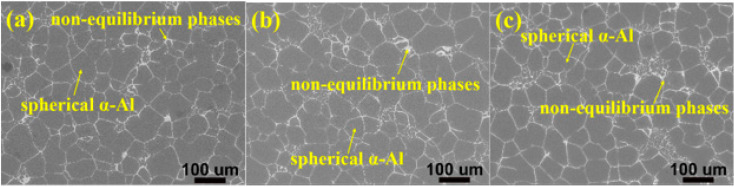
SEM micrographs of the as-cast alloys with different Zn contents: (**a**) 8%, (**b**) 10%, (**c**) 12%.

**Figure 2 materials-15-02873-f002:**
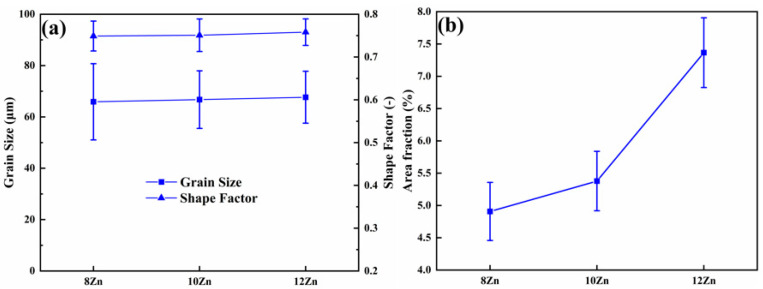
Statistical results of different alloys for (**a**) grain size and shape factor of α-Al and (**b**) area fraction of non-equilibrium phases.

**Figure 3 materials-15-02873-f003:**
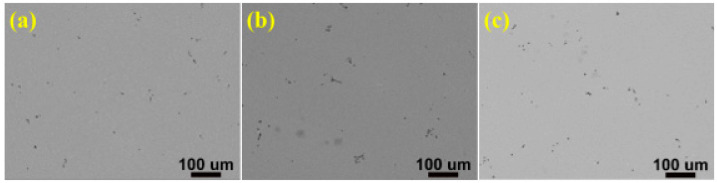
SEM images of as-solution alloys with different Zn contents: (**a**) 8%, (**b**) 10%, (**c**) 12%.

**Figure 4 materials-15-02873-f004:**
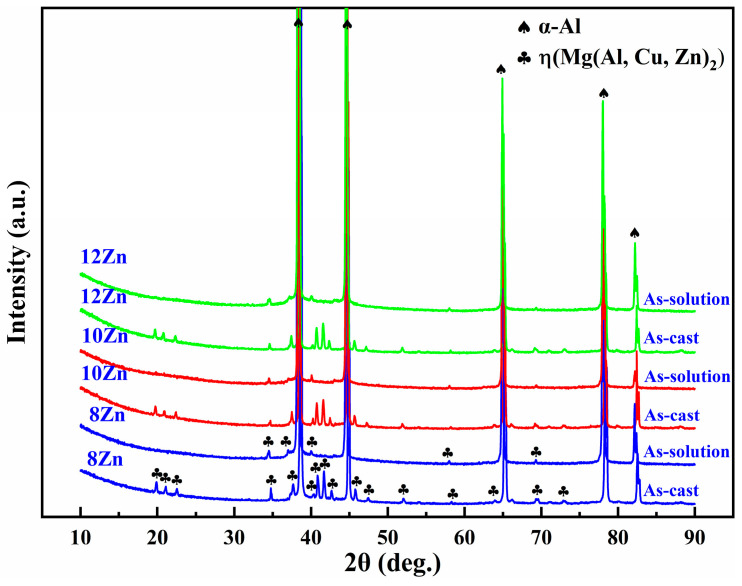
XRD diffraction patterns of as-cast and as-solution alloys with identified phases.

**Figure 5 materials-15-02873-f005:**
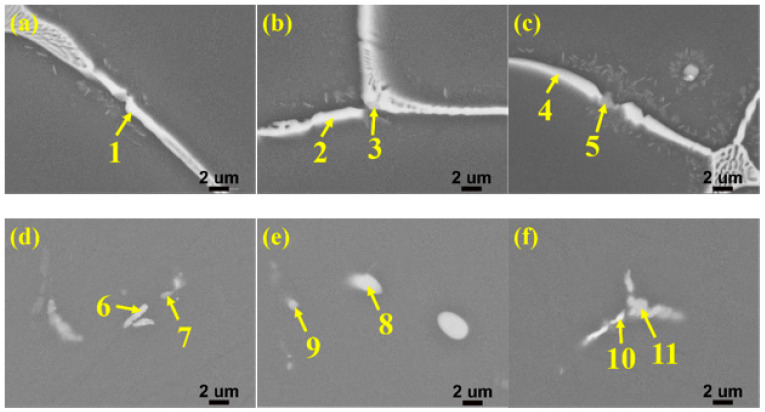
SEM micrographs of as-cast alloys with different Zn contents: (**a**) 8%, (**b**) 10%, (**c**) 12% and as-solution alloys with different Zn contents: (**d**) 8%, (**e**) 10%, (**f**) 12%.

**Figure 6 materials-15-02873-f006:**
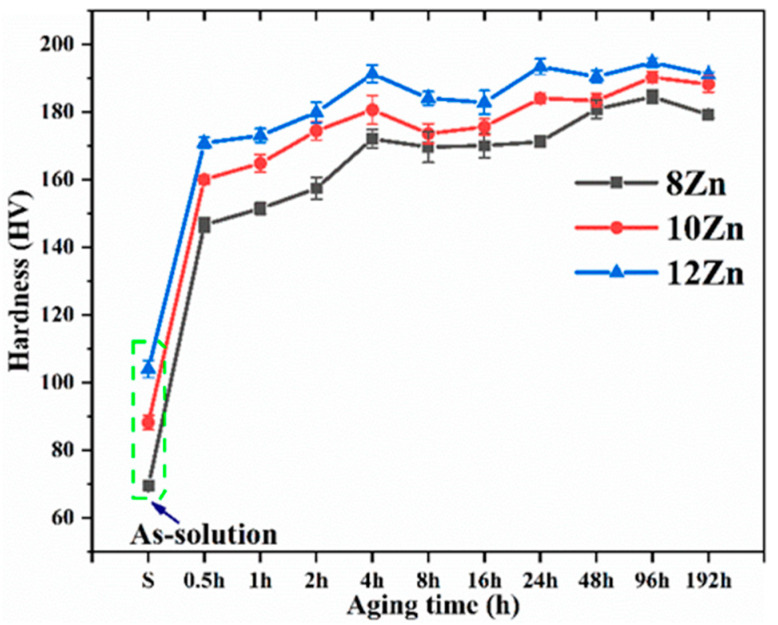
Artificial age hardening curves of the three alloys (470 °C 8 h + 120 °C x h).

**Figure 7 materials-15-02873-f007:**
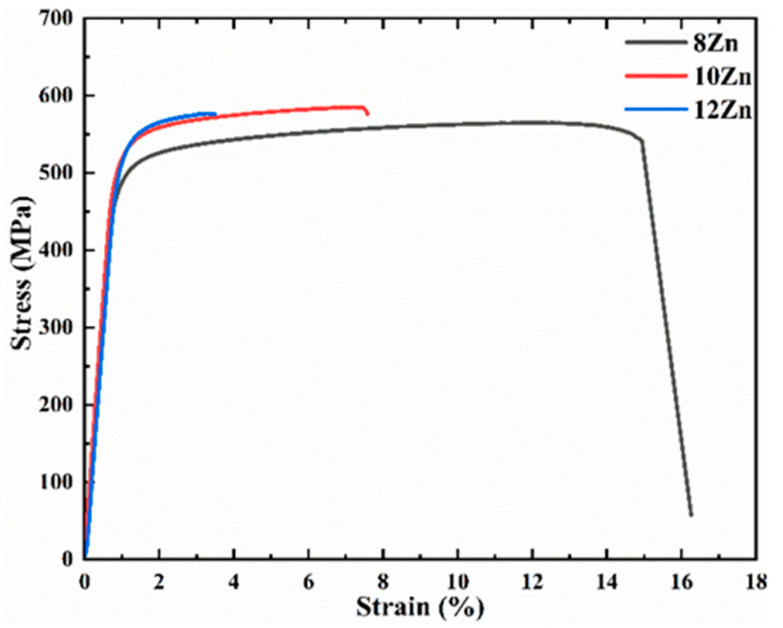
The typical stress-strain curve of the studied alloys in T6 state (470 °C 8 h + 120 °C 24 h).

**Figure 8 materials-15-02873-f008:**
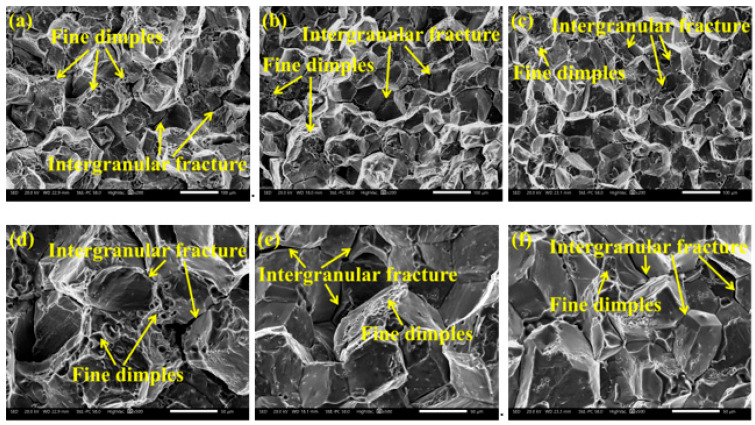
Fracture of T6 state alloys with different Zn contents: (**a**,**d**): 8%, (**b**,**e**): 10%, (**c**,**f**): 12%. (**d**–**f**): the high magnification images of the same sample of (**a**–**c**), respectively.

**Table 1 materials-15-02873-t001:** Nominal composition of the investigated alloys (in wt.%).

Alloys	Zn	Mg	Cu	Ti	Fe	Si	Al
8 Zn alloy	8.22	2.04	1.49	0.094	0.012	0.018	Bal.
10 Zn alloy	10.21	2.01	1.48	0.077	0.011	0.0114	Bal.
12 Zn alloy	13.64	2.14	1.70	0.14	0.014	0.0123	Bal.

**Table 2 materials-15-02873-t002:** EDS results of selected constituent particles highlighted in Figure 5 (in at.%).

Point	Al	Zn	Mg	Cu	Fe	Closest Phase
1	56.97	18.04	18.92	6.06	-	Mg(Al, Cu, Zn)_2_
2	64.36	16.67	14.63	4.34	-	Mg(Al, Cu, Zn)_2_
3	65.19	13.16	8.07	9	4.57	Al_7_Cu_2_Fe
4	65.29	18.86	12.08	3.76	-	Mg(Al, Cu, Zn)_2_
5	86.78	4.22	-	3.13	5.88	Al_7_Cu_2_Fe
6	87.83	3.32	3.07	5.79	-	Mg(Al, Cu, Zn)_2_
7	90.58	2.17	2.65	3.18	1.42	Al_7_Cu_2_Fe
8	61.64	13.31	17.33	7.71	-	Mg(Al, Cu, Zn)_2_
9	85.16	2.16	2.73	6.85	3.1	Al_7_Cu_2_Fe
10	81.68	6.37	2.7	9.26	-	Mg(Al, Cu, Zn)_2_
11	78.33	1.29	1.84	12.27	6.26	Al_7_Cu_2_Fe

## Data Availability

All data are available from the corresponding author on reasonable request.

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
