# Peer review of "Effects of Zn Contents on Microstructure and Mechanical Properties of Semisolid Rheo-Diecasting Al-xZn-2Mg-1.5Cu Alloys"

_materials, 2022, doi:10.3390/ma15082873_

Round 1
Reviewer 1 Report
Very interesting investigation. Bevor publishing, I have some comments which could improve the work.
Abstract:
- L16: After solution heat treatment at 470°C for 8h, most of the Mg(Al, Cu, Zn)2 dissolves into the α-Al matrix, while the Al7Cu2Fe phase keeps undissolved. It is maybe better to replace keeps with remains.
Introduction:
- L48-49: The ultra-high strength of Al-Zn-Mg-Cu alloys was achieved by the precipitation strengthening mechanism. Please reveal more information and cite the following paper, which shows a high UTS of AA7075.
https://doi.org/10.1016/j.matchar.2021.111026
https://doi.org/10.1016/j.msea.2020.140515 - L51 : it should be (MgZn2)
Material and methods
- Please reveal more information about the used SEM
Results
- In figure 2 (a) Shape factor (-)
- In fig. 2 (b) -> it should be. Area fraction (%). See fig 2 (a)
- What is the reason for choosing the aging temperature of 120°C? Please explain that.
Reviewer 2 Report
Dear editor,
This paper experimentally investigates the Effect of Zn contents on microstructure and mechanical properties of semisolid rheo-diecasting Al-xZn-2Mg-1.5Cu alloys. My main concern with the paper is lack of novelty. As the other noted at the end of “Introduction” there are published papers dealing with the effect of Zn content on the microstructure and mechanical properties of Al-Zn-Mg-Cu alloys. None of these papers have been directly cited in the manuscript. Therefore, it seems that the innovation of this paper should be another matter e.g.
- Effect of Zn content on the microstructure and mechanical properties of semisolid rheo-diecast Al-Zn-Mg-Cu alloys
- Effect of Zn content on the heat treatment response of semisolid rheo-diecast Al-Zn-Mg-Cu alloys
Or deeper review on the previous works performed on the same alloy.
However, none of them was considered.
Other important issues are as follows:
- There is no information about semisolid rheo-diecast process in the paper. Given the title of the paper, I thought there was a lively discussion on this topic
- The authors noted that “The evolution of Zn-rich phase during heat treatment was observed, and the age hardening curves of the three alloys was also obtained”. I cannot find the obtained results.
- The reported results are often obvious! The effect of three Zn contents on the microstructure, area fraction of Zn-rich second phases, mechanical properties, etc.
- The authors have discussed the formation of Fe-rich phases in the microstructure. This is while the Fe content in the alloys composition is very low (Fe < 0.0001 wt. %)!
- The fracture surfaces have a “rock candy” appearance which resembles a brittle intergranular fracture. Why?
- There has been no proper discussion on the obtained results or comparing them with the results obtained in other works.
Reviewer 3 Report
Dear authors. Congratulations on a good manuscript that contains many experimental results. However, at the same time, it raises a number of questions.
1. For a more detailed analysis of phase formation, the use of transmission electron microscopy would have been more obvious. Why was this method not used? Well, scanning electron microscopy provides important data on particle distribution and density, but TEM data could have provided more information on the nature of their formation and the GP zones.
2. It is obvious that increasing the Zn content leads to embrittlement of the material, which is shown in the hardness and ultimate tensile strength measurements. Nevertheless, this paper does not make any discussion or conclusions about how these alloys can be used considering the dramatic decrease in ductility.
3. Based on the previous comment, the authors should add a discussion section and talk about the relationship of mechanical properties with the formation of secondary phase particles, their volume fraction, etc., and draw conclusions about the feasibility of using alloys with this concentration of Zn.
Reviewer 4 Report
The material synthesis section is written too general as a kind of report, and not suitable for a scientific paper. It should be rewritten in a concise and concrete form. The experimental details of the tests are not sufficient. Details must be given. Also, the information about the supplier and purity of the starting materials is highly desirable.
Please describe equipment used in the experiment – work development environment / work apparatus should be given – model of equipment (manufacturer, city, country).
Introduction must be discussed along with the novelty of the present approach over the earlier one. Please apply. Also, at the end of the introduction, the authors described the purpose (objectives) of the study, mentioning also the methods used, obviously briefly, but this should be explained in more detail in the methods.
Results and discussion. The discussion presented is poor, in terms of discussing its results and comparing them with the bibliography. I suggest reviewing this part more carefully and discuss further. It is also necessary to critically evaluate new data and do not make hasty conclusions which may lead to misinterpretations.
Conclusions. Please reformulate this section in order to restate the major findings, tell the reader the contribution of this study to the existing literature, state future directions for research, etc.
References must follow the journal guidelines, please check and revise them accordingly.
Improve the scientific rigor of the manuscript.
Finally, I consider that the paper is not proper for publication in the present format and requires large, large revision. Nevertheless, the efforts of performing all the experiments have been significant and I hope that in the near future all the issues will be solved.
Specific comments. Figure 4. Please use Intensity (a.u.) on the y-scale. Table 2. What about the accuracy of the composition of the non-equilibrium phases?
Round 2
Reviewer 2 Report
Thank you for your efforts to improve the quality of the article.Author Response
Thank you for your precious suggestions. I have learned a lot from your suggestions.
Reviewer 3 Report
Dear authors, you did a good job making quality revisions. In this regard, I will recommend your manuscript for publication.
Author Response
Thank you for your precious suggestions. I have learned a lot from your suggestions.
Reviewer 4 Report
The authors have addressed my queries satisfactorily. Before publishing, the quality of figures should be considerable improved. Some descriptors in the plots are very hard to read as they are very small.
Author Response
Thank you for your precious suggestions. I have learned a lot from your suggestions. The quality of figures in the paper has been improved. We have adjusted the resolution of the figures and the size of the descriptors in the plots.